# Improving Cross-Lingual Transfer through Subtree-Aware Word Reordering

**Ofir Arviv**[1] **Dmitry Nikolaev**[2] **Taelin Karidi**[1] **Omri Abend**[1]
[1] Hebrew University of Jerusalem    [2] IMS, University of Stuttgart
`{ofir.arviv,taelin.karidi,omri.abend}@mail.huji.ac.il`
`dnikolaev@fastmail.com`

## Abstract

Despite the impressive growth of the abilities of multilingual language models, such as XLM-R and mT5, it has been shown that they still face difficulties when tackling typologically-distant languages, particularly in the low-resource setting. One obstacle for effective cross-lingual transfer is variability in word-order patterns. It can be potentially mitigated via source- or target-side word reordering, and numerous approaches to reordering have been proposed. However, they rely on language-specific rules, work on the level of POS tags, or only target the main clause, leaving subordinate clauses intact. To address these limitations, we present a new powerful reordering method, defined in terms of Universal Dependencies, that is able to learn fine-grained word-order patterns conditioned on the syntactic context from a small amount of annotated data and can be applied at all levels of the syntactic tree. We conduct experiments on a diverse set of tasks and show that our method consistently outperforms strong baselines over different language pairs and model architectures. This performance advantage holds true in both zero-shot and few-shot scenarios.[1]

## 1 Introduction

Recent multilingual pre-trained language models (LMs), such as mBERT (Devlin et al., 2019), XLM-RoBERTa (Conneau et al., 2020), mBART (Liu et al., 2020b), and mT5 (Xue et al., 2021), have shown impressive cross-lingual ability, enabling effective transfer in a wide range of cross-lingual natural language processing tasks. However, even the most advanced LLMs are not effective when dealing with less-represented languages, as shown by recent studies (Ruder et al., 2023; Asai et al., 2023; Ahuja et al., 2023). Furthermore, annotating sufficient training data in these languages is not a feasible task, and as a result speakers of underrepresented languages are unable to reap the benefits of modern NLP capabilities (Joshi et al., 2020).

Numerous studies have shown that a key challenge for cross-lingual transfer is the divergence in word order between different languages, which often causes a significant drop in performance (Rasooli and Collins, 2017; Wang and Eisner, 2018; Ahmad et al., 2019; Liu et al., 2020a; Ji et al., 2021; Nikolaev and Pado, 2022; Samardžić et al., 2022).[2] This is unsurprising, given the complex and interdependent nature of word-order (e.g., verb-final languages tend to have postpositions instead of prepositions and place relative clauses before nominal phrases that they modify, while SVO and VSO languages prefer prepositions and postposed relative clauses, see Dryer 1992) and the way it is coupled with the presentation of novel information in sentences (Hawkins, 1992). This is especially true for the majority of underrepresented languages, which demonstrate distinct word order preferences from English and other well-resourced languages.

Motivated by this, we present a reordering method applicable to any language pair, which can be efficiently trained even on a small amount of data, is applicable at all levels of the syntactic tree, and is powerful enough to boost the performance of modern multilingual LMs. The method, defined in terms of Universal Dependencies (UD), is based on pairwise constraints regulating the linear order of subtrees that share a common parent, which we term POCs for "pairwise ordering constraints".

We estimate these constraints based on the probability that the two subtree labels will appear in one order or the other when their parent has a given label. Thus, in terms of UD, we expect, e.g., languages that use pre-nominal adjectival modification

---

[1] Code available at `https://github.com/OfirArviv/ud-based-word-reordering`

[2] From a slightly different perspective, this topic has also been actively studied in the machine translation literature (cf. Steinberger, 1994; Chang and Toutanova, 2007; Murthy et al., 2019).

to assign a high probability to `amods`' preceding their headword, while languages with post-nominal adjectival modification are expected to have a high probability for the other direction.

The estimated POCs are fed as constraints to an SMT solver[3] to produce a general reordering algorithm that can affect reordering of all types of syntactic structures. In addition to being effective, POCs are interpretable and provide a detailed characterization of typical word order patterns in different languages, allowing interpretability as to the effect of word order on cross-lingual transfer.

We evaluate our method on three cross-lingual tasks – dependency parsing, task-oriented semantic parsing, and relation classification – in the zero-shot setting. Such setting is practically useful (see, e.g., Ammar et al. 2016; Schuster et al. 2019; Wang et al. 2019; Xu and Koehn 2021 for successful examples of employing ZS learning cross-lingually) and minimizes the risk of introducing confounds into the analysis.

We further evaluate our method in the scarce-data scenario on the semantic parsing task. This scenario is more realistic as in many cases it is feasible to annotate small amounts of data in specific languages (Ruder et al., 2023).

Experiments show that our method consistently presents a noticeable performance gain compared to the baselines over different language pairs and model architectures, both in the zero-shot and few-shot scenario. This suggests that despite recent advances, stronger multilingual models still faces difficulties in handling cross-lingual word order divergences, and that reordering algorithms, such as ours, can provide a much needed boost in performance in low-resource languages.

Additionally, we investigate the relative effectiveness of our reordering algorithm on two types of neural architectures: encoder-decoder (seq2seq) vs. a classification head stacked on top of a pretrained encoder. Our findings show that the encoder-decoder architecture underperforms in cross-lingual transfer and benefits more strongly from reordering, suggesting that it may struggle with projecting patterns over word-order divergences.

The structure of the paper is as follows: Section 2 surveys related work. The proposed approach is introduced in Section 3. Section 4 de-

scribes the setup for our zero-shot and few-shot experiments, the results of which are presented in Section 5. Section 6 investigates the comparative performance of encoder-based and sequence-to-sequence models, and Section 7 concludes the paper.

## 2 Related Work

A major challenge for cross-lingual transfer stems from word-order differences between the source and target language. This challenge has been the subject of many previous works (e.g., Ahmad et al., 2019; Nikolaev and Pado, 2022), and numerous approaches to overcome it have been proposed.

One of the major approaches of this type is reordering, i.e. rearranging the word order in the source sentences to make them more similar to the target ones or vice versa. Early approaches, mainly in phrase-based statistical machine translation, relied on hand-written rules (Collins et al., 2005), while later attempts were made to extract reordering rules automatically using parallel corpora by minimizing the number of crossing word-alignments (Genzel, 2010; Hitschler et al., 2016).

More recent works focusing on reordering relied on statistics of various linguistic properties such as POS-tags (Wang and Eisner, 2016, 2018; Liu et al., 2020a) and syntactic relations (Rasooli and Collins, 2019). Such statistics can be taken from typological datasets such as WALS (Meng et al., 2019) or extracted from large corpora (Aufrant et al., 2016).

Other works proposed to make architectural changes in the models. Thus, Zhang et al. (2017a) incorporated distortion models into attention-based NMT systems, while Chen et al. (2019) proposed learning reordering embeddings as part of Transformer-based translation systems. More recently, Ji et al. (2021) trained a reordering module as a component of a parsing model to improve cross-lingual structured prediction. Meng et al. (2019) suggested changes to the inference mechanism of graph parsers by incorporating target-language-specific constraintsin inference.

Our work is in line with the proposed solutions to source-sentence reordering, namely treebank reordering, which aim to rearrange the word order of source sentences by linearly permuting the nodes of their dependency-parse trees. Aufrant et al. (2016) and Wang and Eisner (2018) suggested permuting existing dependency treebanks to make their surface POS-sequence statistics close to those of

---

[3]An extension of the SAT solver that can, among other things, include mathematical predicates such as $+$ and $<$ in its constraints and assign integer values to variables.

the target language, in order to improve the performance of delexicalized dependency parsers in the zero-shot scenario. While some improvements were reported, these approaches rely on short POS n-grams and do not capture many important patterns.[4] Liu et al. (2020a) proposed a similar method but used a POS-based language model, trained on a target-language corpus, to guide their algorithm. This provided them with the ability to capture more complex statistics, but utilizing black-box learned models renders their method difficult to interpret.

Rasooli and Collins (2019) proposed a reordering algorithm based on UD, specifically, the dominant dependency direction in the target language, leveraging the rich syntactic information the annotation provides. Their method however, leverages only a small part of UD richness, compared to our method.

We note that previous work on treebank reordering usually only evaluated their methods on UD parsing, using delexicalized models or simple manually aligned cross-lingual word embeddings, which limited the scope of the analysis. In this paper, we experiment with two additional tasks that are not reducible to syntactic parsing: relation classification and semantic parsing. We further extend previous work by using modern multilingual LMs and experimenting with different architectures.

## 3 Approach

Given a sentence $s = s_1, s_2, ..., s_n$ in source language $L_s$, we aim to permute the words in it to mimic the word order of a target language $L_t$. Similarly to previous works (Wang and Eisner, 2018; Liu et al., 2020a), we make the assumption that a contiguous subsequence that forms a constituent in the original sentence should remain a contiguous subsequence after reordering, while the inner order of words in it may change. This prevents subtrees from losing their semantic coherence and is also vital when dealing with tasks such as relation extraction (RE), where some of the subsequence must stay intact in order for the annotation to remain valid. Concretely, instead of permuting the words of sentence $s$, we permute the subtrees of its UD parse tree, thus keeping the subsequences of $s$, as

defined by the parse-tree structure, intact.[5]

We define a set of language-specific constraints-based on the notion of *Pairwise Ordering Distributions*, the tendency of words with specific UD labels to be linearly ordered before words with other specific labels, conditioned on the type of subtree they appear in. To implement a reordering algorithm we use these constraints as input to an SMT solver.

### 3.1 Pairwise Ordering Distributions

Let $T(s)$ be the Universal Dependencies parse tree of sentence $s$ in language $L$, and $\pi = (\pi_1, ..., \pi_n)$ the set of all UD labels. We denote the *pairwise ordering distribution* (POD) in language $L$ of two UD nodes with dependency labels $\pi_i, \pi_j$, in a subtree with the root label $\pi_k$ with:

$$P_{\pi_k, \pi_i, \pi_j} = p; p \in [0, 1] \quad (1)$$

where $p$ is the probability of a node with label $\pi_i$ to be linearly ordered before a node with label $\pi_j$, in a subtree with a root of label $\pi_k$. Note that being linearly ordered before a node with index $i$, means having an index of $j < i$, and that the nodes are direct children of the subtree root. We include a copy of the root node in the computation as one of its own children. Thus we can distinguish between a node acting as a representative of its subtree and the same node acting as the head of that subtree.[6]

### 3.2 Pairwise Ordering Constraints and Reordering

Given the pairwise ordering distribution of target language $L_t$, denoted as $dist_{L_t} = P$, we define a set of pairwise constraints based on it. Concretely, for dependency labels $\pi_k, \pi_i, \pi_j$, we define the following constraint:

$$\pi_k : (\pi_i < \pi_j) = \begin{cases} \mathbf{1}, & \text{if } P_{\pi_k, \pi_i, \pi_j} > 0.5 \\ \mathbf{0} & \text{otherwise} \end{cases} \quad (2)$$

where $\pi_k : (\pi_i < \pi_j) = \mathbf{1}$ indicates that a node $n$ with dependency label $\pi_i$ should be linearly ordered before node $n'$ with dependency label $\pi_j$ if they are direct children of a node with label $\pi_k$.

Using these constraints, we recursively reorder the tokens according to the parse tree $T(s)$ in the following way. For each subtree $T_i \in T(s)$ with

---

[4] Aufrant et al. (2016) further experimented with manually crafting permutations rules using typological data on POS sequences from WALS. This approach is less demanding in terms of data but is more labor intensive and does not lead to better performance.

[5] In principle, UD allows for linearly overlapping subtrees, but such cases are rare in the existing treebanks.

[6] An example set of learned POC statistics is given in Appendix C.

UD label $\pi_j$ and children $n_1, n_2, ..., n_m$, with UD labels $n_{1_\pi}, n_{2_\pi}, ..., n_{m_\pi}$:

1. We extract the pairwise constraints that apply to $T_i$ based on the UD labels of its root and children.

2. We feed the pairwise constraints to the SMT solver[7] and use it to compute a legal ordering of the UD labels, i.e. an order that satisfies all the constraints.

3. If there is such an ordering, we reorder the nodes in $T_i$ accordingly. Otherwise, we revert to the original order.

4. We proceed recursively, top-down, for every subtree in $T$, until all of $T(s)$ is reordered to match $dist_{L_t}$.

For example, assuming the constraints nsubj $\rightarrow$ root, obj $\rightarrow$ root, and obl $\rightarrow$ root for the main clause and obl $\rightarrow$ case, corresponding to a typical SOV language, and assuming that the target language does not have determiners,[8] the sentence

*She$_{nsubj}$ put$_{root}$ [the book]$_{obj}$ [on$_{case}$ the table]$_{obl}$*

will be first reordered as

*She$_{nsubj}$ [the book]$_{obj}$ [on$_{case}$ the table]$_{obl}$ **put**$_{root}$*

and then as

*She$_{nsubj}$ [the book]$_{obj}$ [the table **on**$_{case}$]$_{obl}$ put$_{root}$.*

### 3.3 Estimating the Pairwise Ordering Constraints

In this section we describe two possible methods for estimating the POCs of a language, one relying on the availability of a UD corpus in the target language and one relying on the Bible Corpus.

**Using The UD Treebank.** The first method we use to estimate POCs is by extracting them from corresponding empirical PODs in a UD treebank. When there are multiple treebanks per language, we select one of them as a representative treebank. We use v2.10 of the Universal Dependencies dataset, which contains treebanks for over 100 languages.

**Estimating POCs without a Treebank.** While the UD treebank is vast, there are still hundreds of widely spoken languages missing from it. The coverage of our method can be improved by using annotation projection (Agić et al., 2016) on a massively parallel corpus, such as the Bible Corpus (McCarthy et al., 2020). Approximate POCs can then be extracted from the projected UD trees. While we do not experiment with this setting in this work due to resource limitations, we mention it as a promising future work venue, relying on the work done by Rasooli and Collins (2019), which used this approach successfully, to extract UD statistics, and utilize them in their reordering algorithm on top of annotation projection.

## 4 Experimental Setup

We evaluate our reordering algorithm using three tasks – UD parsing, task-oriented semantic parsing, and relation extraction – and over 13 different target languages, with English as the source language.[9] For each task and target language, we compare the performance of a model trained on the vanilla English dataset against that of a model trained on a transformed (reordered) version of the dataset, using the target-language test set in a zero-shot fashion.

We explore two settings: STANDARD, where we reorder the English dataset according to the target language POCs and use it for training, and EN-SEMBLE, where we train our models on both the vanilla and the reordered English datasets. The main motivation for this is that any reordering algorithm is bound to add noise to the data. First, the underlying multilingual LMs were trained on the standard word order of English, and feeding them English sentences in an unnatural word order will likely produce sub-optimal representations. Secondly, reordering algorithms rely on surface statistics, which, while rich, are a product of statistical estimation and thus imperfect. Lastly, the use of hard constrains may not be justified for target languages with highly flexible word-order[10]. The

---

[7] We use Python bindings of the open-source SMT solver Z3 (de Moura and Bjørner, 2008).

[8] Whose relative position for the sake of the example thus can be selected arbitrarily.

[9] We use English as the source language because it has the biggest variety of training sets for different tasks. It has been shown that for some tasks using a source language with a less strict word order, such as Russian or Czech, can lead to improvements (Nikolaev and Pado, 2022), but in practice they are rather minor and do not outweigh the benefits of having access to more corpora.

[10] In the worst-case scenario, given a very flexible order of a particular pair of syntactic elements in the target language, with $P_{\pi_k, \pi_i, \pi_j} \approx 0.51$, our method will select only one or-

ENSEMBLE setting mitigates these issues and improves the "signal to noise ratio" of the approach.

We use the vanilla multilingual models and the reordering algorithm by Rasooli and Collins (2019) as baselines. To the best of our knowledge, Rasooli and Collins proposed the most recent pre-processing reordering algorithm, which also relies on UD annotation. We re-implement the algorithm and use it in the same settings as our approach.

Lastly, we evaluate our method in the scarce-data setting. We additionally train the models fine-tuned on the vanilla and reordered English datasets on a small number of examples in the target language and record their performance. Due to the large amount of experiments required we conduct this experiment using only our method in the context of the semantic-parsing task, which is the most challenging one in our benchmark (Asai et al., 2023; Ruder et al., 2023), on the mT5 model, and in the ENSEMBLE setting.

## 4.1 Estimating the POCs

We estimate the POCs (§3.2) by extracting the empirical distributions from UD treebanks (see §3.3). While this requires the availability of an external data source in the form of a UD treebank in the target language, we argue that for tasks other than UD parsing this is a reasonable setting as the UD corpora are available for a wide variety of languages. Furthermore, we experiment with various treebank sizes, including ones with as few as 1000 sentences. Further experimentation with even smaller treebanks is deferred to future work. Appendix B lists the treebanks used and their sizes.

## 4.2 Evaluation Tasks

In this section, we describe the tasks we use for evaluation, the models we use for performing the tasks, and the datasets for training and evaluation. All datasets, other than the manually annotated UD corpora, are tokenized and parsed using Trankit (Nguyen et al., 2021). Some datasets contain subsequences that must stay intact in order for the annotation to remain valid (e.g., a proper-name sequence such as *The New York Times* may have internal structure but cannot be reordered). In cases where these subsequences are not part of a single subtree, we manually alter the tree to make them

---

dering as the "correct" one. If this ordering contradicts the original English one, it will be both nearly 50% incorrect and highly unnatural for the encoder. Ensembling thus ensures that the effect of estimation errors is bounded.

so. Such cases mostly arise due to parsing errors and are very rare. The hyper-parameters for all the models are given in Appendix D.

### 4.2.1 UD Parsing

**Dataset.** We use v2.10 of the UD dataset. For training, we use the UD English-EWT corpus with the standard splits. For evaluation, we use the PUD corpora of French, German, Korean, Spanish, Thai and Hindi, as well as the Persian-Seraji, Arabic-PADT, and the Irish-TwittIrish treebanks.

We note that our results are not directly comparable to the vanilla baseline because our model has indirect access to a labeled target dataset, which is used to estimate the POCs. This issue is less of a worry in the next tasks, which are not defined based on UD. We further note that we do not use the same dataset for extracting the information about the target language and for testing the method. [11]

**Model.** We use the AllenNLP (Gardner et al., 2018) implementation of the deep biaffine attention graph-based model of Dozat and Manning (2016). We replace the trainable GloVe embeddings and the BiLSTM encoder with XLM-RoBERTa-large (Conneau et al., 2020). Finally, we do not use gold (or any) POS tags. We report the standard labeled and unlabeled attachment scores (LAS and UAS) for evaluation, averaged over 5 runs.

### 4.2.2 Task-oriented Semantic Parsing

**Datasets.** We use the MTOP (Li et al., 2021) and Multilingual TOP (Xia and Monti, 2021) datasets. MTOP covers 6 languages (English, Spanish, French, German, Hindi and Thai) across 11 domains. In our experiments, we use the *decoupled* representation of the dataset, which removes all the text that does not appear in a leaf slot. This representation is less dependent on the word order constraints and thus poses a higher challenge to reordering algorithms. The Multilingual TOP dataset contains examples in English, Italian, and Japanese and is based on the TOP dataset (Gupta et al., 2018). Similarly to the MTOP dataset, this dataset uses the *decoupled* representation. Both datasets are formulated as a seq2seq task. We use the standard splits for training and evaluation.

**Models.** We use two seq2seq models in our evaluation, a pointer-generator network model (Ron-

---

[11]Note that Thai has only one published UD treebank, so for this experiment we split it in two parts, 500 sentences each, for estimating POCs and testing.

gali et al., 2020) and mT5 (Xue et al., 2021). The pointer-generator network was used in previous works on these datasets (Xia and Monti, 2021; Li et al., 2021); it includes XLM-RoBERTa-large (Conneau et al., 2020) as the encoder and an uninitialized Transformer as a decoder. In this method, the target sequence is comprised of *ontology tokens*, such as [IN:SEND_MESSAGE in the MTOP dataset, and *pointer tokens* representing tokens from the source sequence (e.g. ptr0, which represents the first source-side token). When using mT5, we use the actual tokens and not the pointer tokens as mT5 has copy-mechanism built into it, thus enabling the model to utilize it. In both models, we report the standard exact-match (EM) metric, averaged over 10 runs for the pointer-generator model and 5 runs for mT5.

### 4.2.3 Relation Classification

**Datasets.** We use two sets of relation-extraction datasets: (i) TACRED (Zhang et al., 2017b) (TAC) and Translated TACRED (Arviv et al., 2021) (Trans-TAC), and (ii) IndoRE (Nag et al., 2021). TAC is a relation-extraction dataset with over 100K examples in English, covering 41 relation types. The Trans-TAC dataset contains 533 parallel examples sampled from TAC and translated into Russian and Korean. We use the TAC English dataset for training and Trans-TAC for evaluation. As the TAC train split is too large for efficient training, we only use the first 30k examples.

IndoRE (Nag et al., 2021) contains 21K sentences in Indian languages (Bengali, Hindi, and Telugu) plus English, covering 51 relation types. We use the English portion of the dataset for training, and the Hindi and Telugu languages for evaluation.[12]

**Model.** We use the relation-classification part of the LUKE model (Yamada et al., 2020).[13] The model uses two special tokens to represent the head and the tail entities in the text. The text is fed into an encoder, and the task is solved using a linear classifier trained on the concatenated representation of the head and tail entities. For consistency, we use XLM-RoBERTa-large (Conneau et al., 2020) as the encoder. We report the micro-F1 and macro-F1 metrics, averaged over 5 runs.

---

[12]The Bengali UD treebank is extremely small (17 sentences), which makes it impossible to extract high-quality POCs.

[13]https://github.com/studio-ousia/luke

## 5 Results and Discussion

The results on the various tasks, namely UD parsing, semantic parsing, and relation classification, are presented in Tables 1, 2 and 3 respectively. The few-shot experiment results are in Table 4. Standard deviations are reported in Appendix E.

In UD parsing, the ENSEMBLE setting presents noticeable improvements for the languages that are more typologically distant from English (2.3-4.1 LAS points and 1.8-3.5 UAS points), with the exception of Arabic, in which the scores slightly drop. No noticeable effect is observed for structurally closer languages.

In the STANDARD setting, a smaller increase in performance is present for most distant languages, with a decrease in performance for closes ones, Persian and Arabic. This is in agreement with previous work that showed that reordering algorithms are more beneficial when applied to structurally-divergent language pairs (Wang and Eisner, 2018; Rasooli and Collins, 2019). The ENSEMBLE approach, therefore, seems to be essential for a generally applicable reordering algorithm.

The algorithm by Rasooli and Collins (2019) (RC19), in both settings, presents smaller increase in performance for some typologically-distant languages and no noticeable improvements for others, while sometimes harming the results. This suggests that for this task the surface statistics the algorithm uses are not enough and a more fine-grained approach in needed.

In the semantic-parsing task, the reordering algorithm presents substantial improvements for all languages but Italian in the ENSEMBLE setting (2-6.1 increase in exact match), for the RoBERTa based model. Noticeably, the gains are achieved not only for typologically-distant languages but also for languages close to English, such as French. In the MTOP dataset, the ENSEMBLE setting proves bigger gains over the STANDARD for all languages. In Multilingual-TOP, we surprisingly observe the opposite. Given that Japanese in terms of word order is comparable to Hindi and Italian, to French, we tend to attribute this result to the peculiarities of the dataset. This, however, merits further analysis.

When compared to RC19, the proposed algorithm consistently outperforms it, by an average of about 2 points (in the ENSEMBLE setting).

For mT5 we observe increase in performance, in the ENSEMBLE setting, of 2.5 and 5 points in Thai and Hindi, respectively. For the other languages,

| Target | Base | | Ours | | OursE | | RC19 | | RC19E | | OursE LAS gain |
|---|---|---|---|---|---|---|---|---|---|---|---|
| | UAS | LAS | UAS | LAS | UAS | LAS | UAS | LAS | UAS | LAS | |
| Arabic | 67.5 | 52.3 | 65.9 | 50 | 66.5 | 51.2 | 66.9 | 53.2 | 67.6 | **53.7** | -1.1 |
| Spanish | 84.7 | 77.2 | 83.6 | 75.8 | 84.8 | 77.1 | 84.4 | **77.3** | 84.3 | 77.2 | -0.1 |
| German | 86.6 | 80.6 | 83.5 | 77.1 | 86.7 | **80.7** | 85.5 | 78.3 | 85.3 | 78.1 | 0.1 |
| French | 85 | 79.2 | 82 | 76.3 | 84.9 | 79.4 | 84.5 | **79.9** | 84.4 | **79.9** | 0.2 |
| Persian | 66.2 | 52.3 | 51 | 40.1 | 66.4 | 52.7 | 66.4 | **54.1** | 53.7 | 43.1 | 0.4 |
| Korean | 64 | 46.9 | 66.6 | **49.2** | 65.8 | **49.2** | 63.1 | 46.4 | 63.1 | 46.2 | 2.3 |
| Irish | 54.4 | 38.9 | 54.4 | 38.4 | 58.4 | 41.5 | 60 | **42.4** | 59 | 41.6 | 2.6 |
| Thai | 73.8 | 54.5 | 75.5 | 57.7 | 76 | **58** | 73.7 | 52.7 | 72.7 | 51.4 | 3.5 |
| Hindi | 60.4 | 49.8 | 61.1 | 50.9 | 63.9 | **53.9** | 60.7 | 49.9 | 61.4 | 50.9 | 4.1 |

Table 1: The results (averaged over 5 models) of the application of the reordering algorithm to cross-lingual UD parsing. Columns correspond to evaluations settings and score types; rows correspond to evaluation-dataset languages. The best LAS and UAS scores per language are represented in boldface and underlined, respectively. Abbreviations: RC19 – the algorithm by Rasooli and Collins; E – the ENSEMBLE setting.

| Target | Arch | Base | Ours | OursE | RC19 | RC19E |
|---|---|---|---|---|---|---|
| Hi | XLM | 36.9 | 41.5 | **42.9** | 39.3 | 40.4 |
| | mT5 | 22.7 | 24.8 | 27.7 | 21.8 | 24 |
| Th | XLM | 13.5 | 13.8 | 17.9 | 15.9 | 17.3 |
| | mT5 | 31.2 | 31.8 | **33.6** | 31.7 | 32.8 |
| Fr | XLM | 52.9 | 54 | **59** | 51.1 | 55.9 |
| | mT5 | 46.7 | 43.9 | 46.3 | 44.4 | 46.7 |
| Sp | XLM | 55.3 | 55.5 | **59.6** | 54.6 | 57.6 |
| | mT5 | 46.9 | 43.2 | 46.2 | 47 | 47.9 |
| Ge | XLM | 53.9 | 53.7 | **56.4** | 51.3 | 54.7 |
| | mT5 | 39.9 | 38.4 | 40.6 | 37.8 | 41.4 |
| Ja | XLM | 5.5 | **9.7** | 7.5 | 8.5 | 7.9 |
| It | XLM | 54.4 | **54.7** | 54 | 54 | 53.2 |

Table 2: The results (averaged over 10 an 5 models for XLM and mT5 respectively) of the application of the reordering algorithm to MTOP and Multilingual-Top (above and below the horizontal line respectively). Values are exact-match scores. The best score per language is represented in boldface. Abbreviations: XLM – XLM-RoBERTa; RC19 – the algorithm by Rasooli and Collins; E – the ENSEMBLE setting. Language abbreviations: Hi – Hindi, Th – Thai, Fr – French, Sp – Spanish, Ge – German, Ja – Japanese, It – Italian.

we do not observe a strong impact. We note however, that in French and Spanish, there is a slight drop in the score (less then 1 point). When compared to RC19, our method provide larger gains in Hindi and Thai.

In the few-shot scenario, we observe improved performances for all languages and sample sizes. Surprisingly, the improvements hold even when training on a large sample size of 500, indicating that the model is not able to easily adapt to the target word-order.

Lastly, in the relation-classification task, in the ENSEMBLE setting we observe an increase in performance for all languages (2.3-10.4 increase in the Micro and Macro F1 points). In the STANDARD setting, there is a drop in performance for Hindi and Telugu. When compared to RC19, our algorithm outperforms it in the ENSEMBLE setting, by more than 5 points for Korean, but only by 0.5 points for Russian. In Hindi and Telugu, the performance of both algorithms is close, and RC19 does perform better in some cases.

## 6 Comparison between Encoder-with-Classifier-Head and Seq2Seq Models

Past work has shown that the architecture is an important predictor of the ability of a given model to generalize over cross-lingual word-order divergences. For example, Ahmad et al. (2019) showed that models based on self-attention have a better overall cross-lingual transferability to distant languages than those using RNN-based architectures.

One of the dominant trends in recent years in NLP has been using the sequence-to-sequence formulation to solve an increasing variety of tasks (Kale and Rastogi, 2020; Lewis et al., 2020). Despite that, recent studies (Finegan-Dollak et al., 2018; Keysers et al., 2019; Herzig and Berant, 2019) demonstrated that such models fail at compositional generalization, that is, they do not generalize to structures that were not seen at training time. Herzig and Berant (2021) showed that other model architectures can prove advantageous over seq2seq architecture in that regard, but their work was limited to English.

Here, we take the first steps in examining the cross-lingual transfer capabilities of the seq2seq encoder-decoder architecture (S2S) vs. a classification head stacked over an encoder (E+C), focusing on their ability to bridge word-order divergences.

| Target | Base | | Ours | | OursE | | RC19 | | RC19E | |
|---|---|---|---|---|---|---|---|---|---|---|
| | Mic-F1 | Mac-F1 | Mic-F1 | Mac-F1 | Mic-F1 | Mac-F1 | Mic-F1 | Mac-F1 | Mic-F1 | Mac-F1 |
| Korean | 40.3 | 33.4 | 41 | 35.4 | **50** | 43.8 | 48.2 | 43.1 | 44.2 | 38.8 |
| Russian | 62.7 | 58.1 | 66 | 61.7 | **68.1** | 64.2 | 61.1 | 55.8 | 67.6 | 63.8 |
| Hindi | 78.5 | 74.1 | 77.1 | 72.2 | **80.9** | 76.7 | 80 | 75.4 | 80.7 | 77 |
| Telugu | 66.6 | 58.9 | 65.5 | 58.3 | 69.1 | 62 | **70.3** | 61.8 | 69.4 | 62.3 |

Table 3: The results (averaged over 5 models) of the application of the reordering algorithm to Translated Tacred and IndoRE (above and below the horizontal line respectively). Columns correspond to evaluations settings and score types; rows correspond to evaluation-dataset languages. The best Micro-F1 and Macro-F1 scores, per language, are represented in boldface and underlined, respectively. Abbreviations: Mic-F1 – Micro-F1; Mac-F1 – Macro-F1; RC19 – the algorithm by Rasooli and Collins; E – the ENSEMBLE setting.

| Sample size | Hindi | | Thai | | French | | Spanish | | German | |
|---|---|---|---|---|---|---|---|---|---|---|
| | Base | OursE | Base | OursE | Base | OursE | Base | OursE | Base | OursE |
| 100 | 45.2 | 48.3 | 47.6 | 48.4 | 62.3 | 63.1 | 62.9 | 64.2 | 57.8 | 58.7 |
| 300 | 53.3 | 56 | 54.3 | 55.8 | 67 | 67.2 | 67.7 | 68.1 | 61.6 | 62.9 |
| 500 | 56.9 | 58.4 | 58.6 | 59.7 | 67.7 | 68.8 | 70 | 71 | 63.2 | 65.1 |

Table 4: The results (averaged over 5 models) of the application of the reordering algorithm to MTOP in the few-shot scenario, on the mT5 model. Columns correspond to evaluations settings; rows correspond to evaluation-dataset languages. Values are exact-match scores. E – the ENSEMBLE setting.

## 6.1 Experimental Setup

We compare the performance of an E+C model against a S2S one on the task of UD parsing over various target languages. Similar to §4, we train each model on the vanilla English dataset and compare it against a model trained on a version of the dataset reordered using our algorithm. We evaluate the models using the target-language test set in a zero-shot setting.

**Dataset and POCs Estimates.** We use the same UD dataset and POCs as in §4. For the S2S task, we linearize the UD parse tree using the method by Li et al. (2018).

**Models.** For the E+C model we use the deep biaffine attention graph-based model with XLM-RoBERTa-large as the encoder, as in §4.2.1. For S2S model, we use the standard transformer architecture with XLM-RoBERTa-large as the encoder and an uninitialized self-attention stack as the decoder. The hyper-parameters for the models are given in Appendix D.

## 6.2 Results and Discussion

The results for LAS (averaged over 5 runs), normalized by the base parser performance on the English test-set, are presented in Table 5 (UAS follow the same trends. See full scores in Appendix F). The zero-shot performance of the S2S model is subpar compared to the E+C one (less

| Target | Base | | Ours | | OursE | |
|---|---|---|---|---|---|---|
| | E+C | S2S | E+C | S2S | E+C | S2S |
| French | 85.9 | 46.6 | 82.7 | 42.7 | **86.12** | 47.8 |
| German | 87.4 | 54.5 | 83.6 | 56.9 | **87.5** | 67.8 |
| Hindi | 54 | 20.6 | 55.2 | 38.1 | **58.4** | 48.7 |
| Korean | 50.9 | 14.2 | **53.4** | 13.8 | **53.4** | 16.5 |
| Persian | 56.7 | 21.4 | 54.1 | 24.5 | **57.1** | 27.3 |
| Spanish | **83.7** | 63.4 | 82.2 | 59.5 | 83.6 | 64.7 |
| Thai | 59.9 | 28 | 61.1 | 29.5 | **63.7** | 37.9 |

Table 5: LAS results (averaged over 5 models) of the application of the reordering algorithm to cross-lingual UD parsing, normalized by the English test set LAS. Columns correspond to the evaluations settings, and model types; rows correspond to evaluation dataset language. The best scores for the E+C and S2S settings, per language, are represented in boldface and underlined, respectively. Abbreviations: E – the ENSEMBLE setting; E+C – classification head stacked over an encoder architecture; S2S — seq2seq encoder-decoder architecture.

then 50%), despite relying on the same underline multilingual LM. Furthermore, the S2S architecture benefits more strongly from reordering for distant languages (more than twice as much), compared to the E+C one. This suggests that seq-to-sequence architecture may be less effective in handling cross-lingual divergences, specifically word-order divergences, and may gain more from methods such as reordering.

## 7 Conclusion

We presented a novel pre-processing reordering approach that is defined in terms of Universal Dependencies. Experiments on three tasks and numerous architectures and target languages demonstrate that this method is able to boost the performances of modern multilingual LMs both in the zero-shot and few-shot setting. Our key contributions include a new method for reordering sentences based on fine-grained word-order statistics, the Pairwise Ordering Distributions, using an SMT solver to convert the learned constraints into a linear ordering, and a demonstration of the necessity of combining the reordered dataset with the original one (the ENSEMBLE setting) in order to consistently boost performance.

Our results suggest that despite the recent improvements in multilingual models, they still face difficulties in handling cross-lingual word order divergences, and that reordering algorithms, such as ours, can provide a much needed boost in performance in low-resource languages. This result holds even in the few-shot scenario, when the model is trained on few hundred examples, underscoring the difficulty of models to adapt to varying word orders, as well as the need for more typologically diverse data, additional inductive bias at the training time, or a pre-processing approach such as ours to be more effective. Furthermore, our experiments suggest that seq2seq encoder-decoder architectures may suffer from these difficulties to a bigger extent than more traditional modular ones.

Future work will include, firstly, addressing the limitations of the proposed approach in order to make it less language-pair dependent and reduce the computational and storage overhead, and secondly, leveraging the POCs in order to compute the word-order distance between languages in a rich, rigorous corpus-based way,[14] and to more precisely predict when the reordering algorithm will be beneficial as well as to provide a fine-grained analysis of the connection between word order and cross-lingual performance, in line with Nikolaev et al. (2020).

## Limitations

There are several limitation to our work. First, as shown in the experiment results, for some tasks, reordering, even with ensembling, is not beneficial

---

[14]WALS, for example, only provides a single categorical label for "dominant word order".

for closely-related languages. Secondly, as this is a pre-processing algorithm, it incurs time and computation overhead for tokenization, parsing, and reordering of the source dataset. Most importantly, if one wishes to train a model on several word order patterns, it will be necessary to apply the algorithm repeatedly to create a new dataset for each pattern, increasing the overall size of the dataset. Furthermore, extracting the POCs requires external resources, either UD corpora in the target language or a multi-parallel corpus for annotation projection. More technical limitations of the reordering algorithm are described in Appendix A.

## Acknowledgements

This work was supported in part by the Israel Science Foundation (grant no. 2424/21).

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

# A Shortcomings of the Reordering Algorithm

We note a couple of possible shortcomings to this approach. First, while a pair of constraints cannot be straightforwardly contradictory (both values

cannot be set to 1), it can be uninformative (both values set to 0) when not enough label-ordering data is presented in the training treebank, which means that the ordering of the corresponding nodes is not subject to any constraint.

Moreover, it is possible to encounter loops or transitivity conflicts when joining different constraints, which makes it a priori impossible for the solver to satisfy them. To alleviate this problem, for each subtree we aim to reorder, we only consider the constraints that are relevant to it. For example, if the subtree does not contain any token with label $nmod$, we discard all the constraints which include this label, such as $amod : (nmod < amod)$. This, together with the tendency of languages to have a preferred ordering to their constituent elements, makes it so that only a small percentage of subtrees cannot be ordered.

Last, sparsity issues may prevent some constraints from being statistically justified, and rounding the constraints to a hard 0 or 1 may result in information loss and thus be detrimental. For example, if our pairwise distributions are $P_{nmod,nmod,amod} = 0.51$ and $P_{nmod,amod,nmod} = 0.49$, deriving a constraint of $nmod : (nmod < amod)$ may not be warranted. This may also happens in the case of a highly flexible order of a particular pair of syntactic elements in the target language. If this ordering contradicts the original English one, it will be both nearly 50% incorrect and highly unnatural for the encoder. This is, however, partially mitigated by the ENSEMBLE\$ method. For the purposes of this work we do not distinguish between POCs according to their statistical validity and defer this question to future work.

## B  Estimating the POCs

We estimate the POCs (§3.2) by extracting the empirical distributions from UD treebanks. The UD treebanks used are reported in Table 6.

## C  Example of Learned Distributions

Here are the statistics of the pairwise ordering of main elements of the matrix clause[15] learned on the Irish-IDT treebank:

- acl vs. advcl
    - acl->advcl: 10
    - advcl->acl: 4

[15]Elements that are directly under the `root` node.

| Language | UD Treebank | Size |
|---|---|---|
| **French** | French-GSD | 14450 |
| **Spanish** | Spanish-GSD | 14187 |
| **German** | German-GSD | 13814 |
| **Italian** | Italian-ISDT | 13121 |
| **Russian** | Russian-GSD | 3850 |
| **Irish** | Irish-IDT | 4005 |
| **Arabic** | Arabic-PUD | 1000 |
| **Korean** | Korean-GSD | 4400 |
| **Japanese** | Japanese-GSD | 7050 |
| **Thai** | Thai-PUD | 1000 |
| **Persian** | Persian-PerDT | 26196 |
| **Hindi** | Hindi-HDTB | 13306 |
| **Telugu** | Telugu-MTG | 1051 |

Table 6: The UD treebanks used to estimate the POCs. Each row corresponds to a language-treebank pair. Treebank size is measured in sentence counts.

| Target | Base | | Ours | | OursE | |
|---|---|---|---|---|---|---|
| | UAS | LAS | UAS | LAS | UAS | LAS |
| French | 44.3 | 39.1 | 41.1 | 35.8 | 45.4 | 40.1 |
| German | 51.2 | 45.7 | 53.7 | 47.7 | 62.4 | 56.8 |
| Spanish | 59.7 | 53.1 | 56.5 | 49.9 | 60.6 | 54.2 |
| Korean | 26.4 | 11.9 | 26.6 | 11.6 | 28.3 | 13.8 |
| Persian | 27.3 | 17.9 | 29 | 20.5 | 32.4 | 22.9 |
| Hindi | 26.9 | 17.3 | 41.1 | 31.9 | 50.4 | 40.8 |
| Thai | 35.6 | 23.5 | 37.7 | 24.7 | 45.5 | 31.8 |

Table 7: The results (averaged over 5 models) of the application of the reordering algorithm to cross-lingual s2s UD parsing. Columns correspond to evaluations settings and score types; rows correspond to evaluation-dataset languages. Abbreviations: RC19 – the algorithm by Rasooli and Collins; E – the ENSEMBLE setting.

- acl vs. advmod
    - advmod->acl: 6
    - acl->advmod: 1
- acl vs. amod
    - amod->acl: 29
    - acl->amod: 1
- acl vs. appos
    - appos->acl: 3
    - acl->appos: 4
- acl vs. nsubj
    - nsubj->acl: 35
    - acl->nsubj: 8
- acl vs. obj
    - obj->acl: 3
- acl vs. obl
    - obl->acl: 14

| Target | Base | | Ours | | OursE | |
|--------|------|-----|------|-----|-------|-----|
| | UAS | LAS | UAS | LAS | UAS | LAS |
| French | 1 | 1.1 | 0.5 | 0.6 | 1.6 | 1.7 |
| German | 1.2 | 1.2 | 0.4 | 0.3 | 0.4 | 0.3 |
| Spanish | 0.7 | 0.6 | 0.4 | 0.4 | 0.2 | 0.3 |
| Korean | 0.5 | 0.5 | 0.7 | 0.6 | 0.4 | 0.5 |
| Persian | 0.5 | 0.4 | 0.7 | 0.6 | 0.4 | 0.4 |
| Hindi | 0.7 | 0.5 | 0.6 | 0.5 | 0.7 | 0.7 |
| Thai | 0.9 | 0.7 | 2.8 | 2.6 | 1.2 | 2.2 |

Table 8: The standard deviation of the results (averaged over 5 models) of the application of the reordering algorithm to cross-lingual s2s UD parsing. Columns correspond to evaluations settings and score types; rows correspond to evaluation-dataset languages. Abbreviations: RC19 – the algorithm by Rasooli and Collins; E – the ENSEMBLE setting.

  – acl->obl: 4
- acl vs. root
  – root->acl: 144
- advcl vs. advcl
  – advcl->advcl: 62
- advcl vs. advmod
  – advmod->advcl: 72
  – advcl->advmod: 22
- advcl vs. amod
  – amod->advcl: 15
  – advcl->amod: 5
- advcl vs. appos
  – advcl->appos: 2
- advcl vs. nsubj
  – advcl->nsubj: 132
  – nsubj->advcl: 320
- advcl vs. obj
  – obj->advcl: 128
  – advcl->obj: 48
- advcl vs. obl
  – obl->advcl: 339
  – advcl->obl: 189
- advcl vs. root
  – root->advcl: 513
  – advcl->root: 243
- advmod vs. amod
  – advmod->amod: 3
  – amod->advmod: 1
- advmod vs. nsubj
  – nsubj->advmod: 337
  – advmod->nsubj: 83
- advmod vs. obj

  – obj->advmod: 71
  – advmod->obj: 77
- advmod vs. obl
  – obl->advmod: 231
  – advmod->obl: 326
- advmod vs. root
  – advmod->root: 111
  – root->advmod: 486
- amod vs. appos
  – amod->appos: 2
- amod vs. nsubj
  – nsubj->amod: 13
  – amod->nsubj: 41
- amod vs. obj
  – obj->amod: 3
- amod vs. obl
  – obl->amod: 7
  – amod->obl: 11
- amod vs. root
  – root->amod: 133
  – amod->root: 6
- appos vs. nsubj
  – nsubj->appos: 5
  – appos->nsubj: 4
- appos vs. obl
  – obl->appos: 3
- appos vs. root
  – root->appos: 32
- nsubj vs. obj
  – nsubj->obj: 469
  – obj->nsubj: 2
- nsubj vs. obl
  – nsubj->obl: 1699
  – obl->nsubj: 314
- nsubj vs. root
  – root->nsubj: 2423
  – nsubj->root: 60
- obj vs. obl
  – obj->obl: 755
  – obl->obj: 219
- obj vs. root
  – root->obj: 907
  – obj->root: 13
- obl vs. root
  – root->obl: 2566
  – obl->root: 425

As expected, Irish behaves as a strict head-initial language: `root` overwhelmingly precedes all other constituents, including subordinate clauses, and modifier subordinate clauses (`acl, advcl`) follow nominal clause participants (`nsubj, obj, obl`). Adjectival modifiers, however, mostly precede nominal elements; this may be due to the fact that some frequent pronominal adjectives, such as *uile* 'all' and *gach* 'every' do not follow the general rule and precede the nouns they modify.

The position of adverbial modifiers is not restricted by the grammar, and it may be noted that it generally follows nominal subjects but as often as not precedes direct objects, and in 3/5 of cases precedes obliques, which suggests the general order `root → nsubj → advmod/obj → obl`.

## D    Models Hyperparameters

The hyperparameters of the UD parser are given in Table 9. For the seq2seq pointer-generator network model – in Table 10, for mT5 – in Table 11, and for LUKE relation classification model – in Table 12.

For the UD Seq2Seq parser, we use the same hyperparameters as for the seq2seq pointer-generator network model, with the following exceptions: we train for only 50 epochs and set the learning rate to 1e–5 for both the encoder and decoder.

## E    Standard Deviations

The standard deviations of the results of the experiments in UD parsing, semantic parsing, and relation classification are presented in Tables 14, 13, and 15 respectively.

## F    UD Seq2Seq Model Performances

The full results (averaged over 5 models) of the s2s model in UD parsing, are presented in Table 7. The standard deviations are in Table 8.

| | Hyperparameter | Value |
|---|---|---|
| **Dataset Reader** | input max tokens | 100 |
| **Encoder** | type | xlm-roberta-large |
| | train_parameters | true |
| **Model (General)** | type: | biaffine_parser |
| | arc_representation_dim | 500 |
| | tag_representation_dim | 100 |
| | dropout | 0.1 |
| | input_dropout | 0.3 |
| **Training** | num_epochs | 100 |
| | patience | 10 |
| | grad_norm | 5.0 |
| **Optimizer** | type | huggingface_adamw |
| | lr | 1e-5 |
| | weight_decay | 0.01 |

Table 9: Hyperparameters for the biaffine UD parser.

| | Hyperparameter | Value |
|---|---|---|
| **Dataset Reader** | input max tokens | 100 |
| **Encoder** | type | xlm-roberta-large |
| | train_parameters | true |
| **Decoder** | type | stacked_self_attention |
| | num_layers | 4 |
| | num_attention_heads | 8 |
| | decoding_dim | 1024 |
| | target_embedding_dim | 1024 |
| | feedforward_hidden_dim | 512 |
| | pointers_copy_mechanisem (if using pointers)} | BilinearAttention |
| **Model (General)** | label_smoothing_ratio | 0.1 |
| | beam_search_beam_size | 4 |
| | beam_search_max_steps | 100 |
| **Training** | num_epochs | 100 |
| | patience | None |
| | grad_norm | 5.0 |
| **Optimizer** | type | huggingface_adamw |
| | lr | encoder: 1e-3 decoder: 1e-5 |
| | weight_decay | 0.01 |
| | learning_rate_scheduler | slanted_triangular |
| | gradual_unfreezing | true |

Table 10: Hyperparameters for the pointer network generator seq2seq model.

| | Hyperparameter | Value |
|---|---|---|
| **Dataset Reader** | input max tokens | 100 |
| **Model (General)** | type: | google/mt5-base |
| | label_smoothing_ratio | 0.1 |
| | beam_search_beam_size | 4 |
| | beam_search_max_steps | 100 |
| **Training** | num_epochs | 50 |
| | patience | None |
| | grad_norm | 5.0 |
| **Optimizer** | type | huggingface_adamw |
| | lr | 1e-5 |
| | weight_decay | 0.01 |

Table 11: Hyperparameters for mT5.

| | Hyperparameter | Value |
|---|---|---|
| **Encoder** | type | xlm-roberta-large |
| | train_parameters | true |
| **Seq2vec Encoder** | type: | bert_pooler |
| | dropout | 0.1 |
| **Training** | num_epochs | 100 |
| | patience | 5 |
| | grad_norm | 5.0 |
| **Optimizer** | type | huggingface_adamw |
| | lr | 1e-5 |
| | weight_decay | 0.01 |

Table 12: Hyperparameters for the LUKE relation classification model.

| Target | Arch | Base | Ours | OursE | RC19 | RC19E |
|---|---|---|---|---|---|---|
| Hi | XLM | 1.5 | 1.1 | 1.2 | 1.1 | 0.8 |
| | mT5 | 1 | 0.8 | 1 | 1.3 | 0.9 |
| Th | XLM | 1.8 | 1.9 | 2.7 | 1.6 | 2.1 |
| | mT5 | 0.5 | 1.1 | 0.8 | 0.8 | 0.2 |
| Fr | XLM | 1.6 | 1.9 | 1.3 | 1.1 | 0.8 |
| | mT5 | 0.9 | 0.9 | 0.5 | 0.5 | 1.8 |
| Sp | XLM | 1.5 | 1.1 | 1.2 | 1.3 | 0.9 |
| | mT5 | 0.3 | 0.6 | 0.9 | 1 | 0.3 |
| Ge | XLM | 0.9 | 0.4 | 0.8 | 1.1 | 1 |
| | mT5 | 0.4 | 0.6 | 0.5 | 0.6 | 0.3 |
| Ja | XLM | 1.7 | 1.4 | 2 | 1 | 1.1 |
| It | XLM | 0.4 | 0.7 | 1.1 | 0.5 | 0.8 |

Table 13: Standard deviation (averaged over 10 an 5 models for XLM and mT5 respectively) of the results of the application of the reordering algorithm to MTOP and Multilingual-Top (above and below the horizontal line respectively). For the mT5 model, we report results only for the MTOP dataset, due to time constraints. Columns correspond to evaluations settings; rows correspond to evaluation-dataset languages and model types. Values are exact-match scores. Abbreviations: XLM – XLM-RoBERTa; RC19 – the algorithm by Rasooli and Collins; E – the ENSEMBLE setting. Language abbreviations: Hi – Hindi, Th – Thai, Fr – French, Sp – Spanish, Ge – German, Ja – Japanese, It – Italian.

| Target | Base | | Ours | | Ours-E | | Ras | | Ras-E | |
|---|---|---|---|---|---|---|---|---|---|---|
| | UAS | LAS | UAS | LAS | UAS | LAS | UAS | LAS | UAS | LAS |
| French | 0.1 | 0.1 | 0.9 | 1 | 0.2 | 0.2 | 0.2 | 0.3 | 0.2 | 0.2 |
| German | 0.3 | 0.4 | 0.7 | 0.6 | 0.1 | 0.3 | 0.3 | 0.4 | 0.3 | 0.2 |
| Hindi | 0.9 | 1.1 | 0.9 | 0.7 | 0.6 | 0.5 | 1 | 1.1 | 1.1 | 1.2 |
| Korean | 1 | 0.5 | 1 | 0.5 | 0.8 | 0.4 | 1 | 0.4 | 1.3 | 0.5 |
| Persian | 0.6 | 0.7 | 0.8 | 0.5 | 0.8 | 0.5 | 1 | 0.7 | 0.5 | 1.1 |
| Spanish | 0.1 | 0.1 | 0.6 | 0.5 | 0.1 | 0.2 | 0.2 | 0.2 | 0.4 | 0.4 |
| Thai | 1.8 | 1.4 | 0.9 | 0.5 | 0.3 | 0.3 | 1.3 | 1.6 | 1.4 | 3.1 |
| Irish | 2.1 | 1.4 | 1.2 | 0.9 | 1.8 | 1.4 | 0.7 | 0.8 | 0.2 | 0.8 |
| Arabic | 0.6 | 0.8 | 0.7 | 1 | 0.5 | 0.7 | 0.3 | 1.1 | 0.2 | 0.4 |

Table 14: Standard deviations (over 5 models) of the results of the application of the reordering algorithm to cross-lingual UD parsing. Columns correspond to evaluations settings and score types; rows correspond to evaluation-dataset languages. Abbreviations: Ras – the algorithm by Rasooli and Collins; E – the ENSEMBLE setting.

| Target | Base | | Ours | | OursE | | RC19 | | RC19E | |
|---|---|---|---|---|---|---|---|---|---|---|
| | Mic-F1 | Mac-F1 | Mic-F1 | Mac-F1 | Mic-F1 | Mac-F1 | Mic-F1 | Mac-F1 | Mic-F1 | Mac-F1 |
| Korean | 5.3 | 6.4 | 3.4 | 3.6 | 5.3 | 5.9 | 5.4 | 5.8 | 8.4 | 9.1 |
| Russian | 3 | 4 | 2.5 | 3.1 | 3.1 | 3.5 | 4.8 | 6.5 | 2.2 | 2.8 |
| Hindi | 1.6 | 2.5 | 2 | 2.7 | 1 | 1.5 | 0.8 | 1.1 | 1.5 | 1.6 |
| Telugu | 3.8 | 4.1 | 3.3 | 1.7 | 3.1 | 2.3 | 1.8 | 2.7 | 4.5 | 3.8 |

Table 15: The standard deviations of the results (averaged over 5 models) of the application of the reordering algorithm to Translated Tacred and IndoRE (above and below the horizontal line respectively). Columns correspond to evaluations settings and score types; rows correspond to evaluation-dataset languages. Abbreviations: Mic-F1 – Micro-F1; Mac-F1 – Macro-F1; RC19 – the algorithm by Rasooli and Collins; E – the ENSEMBLE setting.

| Target | Sample Size | Base | OursE |
|--------|-------------|------|-------|
| Hi | 100 | 0.9 | 1 |
|    | 300 | 0.9 | 0.9 |
|    | 500 | 1 | 1 |
| Th | 100 | 1.4 | 0.6 |
|    | 300 | 1.7 | 0.7 |
|    | 500 | 1.3 | 0.9 |
| Fr | 100 | 1 | 1.2 |
|    | 300 | 0.7 | 0.8 |
|    | 300 | 0.5 | 0.5 |
| Sp | 100 | 0.2 | 0.9 |
|    | 300 | 0.5 | 0.8 |
|    | 500 | 0.5 | 0.6 |
| Ge | 100 | 0.8 | 1.1 |
|    | 300 | 0.5 | 0.8 |
|    | 500 | 0.8 | 0.9 |

Table 16: The standard deviations of the results (averaged over 5 models) of the application of the reordering algorithm to MTOP in the few-shot scenario. Columns correspond to evaluations settings; rows correspond to evaluation-dataset languages. Values are exact-match scores. Abbreviations: E – the ENSEMBLE setting. Language abbreviations: Hi – Hindi, Th – Thai, Fr – French, Sp – Spanish, Ge – German.