# OpenReview forum: "Improving Cross-lingual Transfer through Subtree-aware Word Reordering"
_EMNLP/2023/Conference — EMNLP 2023 Findings_

### Official Review · Reviewer_gucq · 2023-07-25

**Typos Grammar Style And Presentation Improvements:** 1.	Line 265
**Soundness:** 3

**Excitement:**

3: Ambivalent: It has merits (e.g., it reports state-of-the-art results, the idea is nice), but there are key weaknesses (e.g., it describes incremental work), and it can significantly benefit from another round of revision. However, I won't object to accepting it if my co-reviewers champion it.

**Missing References:**

n/a

**Paper Topic And Main Contributions:**

This paper presents a source-sentence reordering method to narrow the gap between typologically-distant languages in cross-lingual transfer. They learn reordering rules based on Universal Dependencies and apply them at all levels of the syntactic tree. Extensive experiments show the effectiveness of the proposed approach in enhancing cross-lingual transfer, particularly in the context of low-resource languages.

**Questions For The Authors:**

n/a

**Reasons To Accept:**

1.	Extensive experiments on different tasks (UD paring, relation classification, and semantic parsing) and settings (zero-shot and few-shot), which show the necessity of reordering.
2.	Detailed analyses on different architectures.


**Reasons To Reject:**

1.	Insufficient comparison with related works.
First, the differences with the most related work (Rasooli and Collins, 2019) should be clearer. Their approach is reported to leverage rich syntactic information, as stated on line 190, which is controversial with the expression “superficial statistics” in the line 192. Second, it is insufficient to solely compare the proposed methodology with that of Rasooli and Collins (2019). It would be prudent to also consider similar work, such as that of Liu et al. (2020a).

2.	The effectiveness of these two settings (STANDARD and ENSEMBLE) varies depending on the tasks and languages. As shown in table 5, ENSEMBLE outperforms STANDARD for the majority of languages, with the notable exceptions of Japanese and Italian. A similar phenomenon is observed for Thai and Irish, as indicated in Table 2, warranting further investigation.


**Reproducibility:**

4: Could mostly reproduce the results, but there may be some variation because of sample variance or minor variations in their interpretation of the protocol or method.

**Reviewer Confidence:**

4: Quite sure. I tried to check the important points carefully. It's unlikely, though conceivable, that I missed something that should affect my ratings.

---

> ### Author Rebuttal · Authors · 2023-08-28
>
> We thank the reviewer for their review and for the suggestions for improvement and we will incorporate them.
>
>
>
> * **[R1]** What we meant is that while Rasooli and Collins (2019) leveraged the rich syntactic information of Universal Dependencies, they only used a very small part of the information UD provides (although they did put it to great use), in contrast to our work. The “richness” remark refers to UD. We will clarify this.
>
>     About the need to compare with more methods - this was not done for a couple of reasons: first, other methods are not easily reproducible. Specifically, comparing with Liu et al. (2020a) would have required training POS-based language models, and then reproducing their experiments to make sure we managed to train comparable models. This is in itself a big task. We did not find any released code or models available online.
>
>
>     We also note that we needed to create our own implementation of Rasooli and Collins (2019) method to test it.
>
>
>     We plan to release our method (and our implementation of Rasooli and Collins (2019)), to facilitate future replications.
>
>
>     Secondly, the amount of resources required for this is too high - a proper comparison would have required training of hundreds of additional models, and our study being very computationally intensive as it is, we needed to choose our experiments very carefully.
>
>
>     Lastly, to our knowledge Rasooli and Collins (2019) is the most comparable method to ours (being pre-processing-based, interpretable, and based on UD), and it produces strong results.
>
> * **[R2]** For Japanese and Italian in the Multilingual-TOP dataset, we suspect it is due to the peculiarities of the dataset, as this is the only dataset where we see this effect. Regarding Irish and Arabic in Table 2, this happens only for Rasooli and Collins’s (2019) method. In general, we agree that this (and understanding exactly why the ENSEMBLE is better) is an interesting avenue to check, but due to the amount of work already done, it was outside of scope for this work.
> * [**Presentation Improvements 1[** We will add average gains per language class to Table 2, as suggested.
> * [**Presentation Improvements 2]** “Additionally, it may be beneficial to employ the average reordering time per sentence as a quantitative measure of structural distance” — it is hard to measure this properly as for each subtree it happens that either (i) we get an output from the SMT solver (using which we can apply some kind of index-permutation distance to quantify the local WO distance) or (ii) SMT fails to solve the subtree, which makes the estimation of the structural distance impossible. As we do not know the distribution of subtrees that fail to solve, using only solved subtrees to measure the distance can be arbitrarily biased. It is possible, however, to measure WO differences between languages using all learned pairwise constraints. We regard this as an avenue for future work.

---

### Official Review · Reviewer_YA2U · 2023-07-26

**Soundness:** 4

**Excitement:**

4: Strong: This paper deepens the understanding of some phenomenon or lowers the barriers to an existing research direction.

**Missing References:**

Joshi et al., 2020. The State and Fate of Linguistic Diversity and Inclusion in the NLP World
- Could be useful for the case you're making in the introduction

**Paper Topic And Main Contributions:**

The paper presents an interesting approach to reordering for multilingual NLP, which instead of reordering the words of sentence, it reorders the sub-trees of its universal dependencies parse tree. The proposed method is shown to outperform previous approaches both in zero- and few-shots settings.

**Questions For The Authors:**

[1] l. 477 What about Irish? There seems to be an advantage of your method there.

[2] ll. 478-479 "No noticeable effect is observed for structurally closer languages." However there is once again a decrease in performance in Arabic, and a terrible drop in Persian!

[3] ll. 512-513 What do you mean by significant? Did you test it with some statistical test? One option could be the McNemar test on paired nominal data. It is true that the difference is numerically small, but statistical significance (p-values) depend on the variance of the samples. Very small differences can be statistically significant with large sample sizes.

[4] ll. 528-529 How do you explain the drop in performance in Hindi and Telugu? It's in contrast with your previous observation, where more typologically distant languages benefited the most from your approach.

**Reasons To Accept:**

[1] The proposed method is simple, effective, and particularly suited to post-hoc interpretability.

[2] The paper considers both few-shot and zero-shot experiments.

[3] The "Related work" section is very informative and clear. Personally, I do not work on machine translation, and I really enjoyed quality of the overview.

[4] Interestingly, the proposed method is associated with a higher performance gain in the seq2seq setting, highlighting a difficulty of encoder-decoder architectures in dealing with variable word order patterns.

**Reasons To Reject:**

[1] I think that the study could greatly benefit from an objective quantification of the typological / word-order distance between the languages considered. One of the point I found most interesting about this work is that there seems to be an increase in performance for languages that are distant from English (e.g., Japanese, Hindi), and a decrease in performance (sometimes) for languages that are close to English (e.g., Spanish, German, especially with mT5). It would be great to assess this trend more formally, with a metric of typological similarity (e.g., using WALS).

[2] The authors state that their approach is suited for interpretability; however, the way POCs can be interpreted is never truly addressed in the body of the paper. It would have been a very nice follow-up analysis/appendix.

**Reproducibility:**

5: Could easily reproduce the results.

**Reviewer Confidence:**

2: Willing to defend my evaluation, but it is fairly likely that I missed some details, didn't understand some central points, or can't be sure about the novelty of the work.

**Typos Grammar Style And Presentation Improvements:**

[1] Highlighting the best results per condition in bold in the tables would increase readability a lot!

[2] Why is table 2 discussed before table 1?

---

> ### Author Rebuttal · Authors · 2023-08-28
>
> We thank the reviewer for their review, for the reference and for the suggestions for improvement and we will incorporate them.
>
>
>
> * **[R1]** We regard this as an important avenue for future work. Furthermore, while not mentioned in the paper, the estimates can be used in order to compute the word-order distance between languages in a rigorous corpus-based way, while WALS only provides a single categorical label for “dominant word order”.
> * **[R2]** Again, we view this as an important avenue for future work. Time permitting, we will provide an appendix with an overview of learned constraints for some target language.
> * **[Q1]** If you are referring to the fact that we wrote “No noticeable effect is observed for structurally closer languages” while Irish did gain from our ENSEMBLE method - this is because we consider Irish a distant language. Typologically, Irish is not much closer to English than Arabic (in addition to being a VSO language and lacking most of Standard Average European features, it has neither infinitives nor the verb “to have”, its verbal system is heavily reliant on the use of nominalised verb forms, it has a complex system of morphophonological alternations known as “mutations”, it employs conjugated prepositions to index prenominal objects, etc.).
> * **[Q2]** These lines refer to the ENSEMBLE setting (see line 472), in which Persian has a very slight increase.  In line 476 we state “with the exception of Arabic”. In line 478-481 we refer to the STANDARD and state the decrease in performances, but we should indeed be clearer there and we will fix that.
> * **[Q3]** “Significant” here refers to the strength of the effect, not its statistical significance; we will rephrase the passage. Generally, we tried to make the results more robust by running the experiments and examining and reporting standard deviations, but with only 5 data points proper statistical significance testing is not really possible (and running more than 5 was prohibitively costly).
> * **[Q4]** They still benefit from the ENSEMBLE approach. As to why in the STANDARD approach there was a drop in performance, this can be because of the idiosyncrasies of the RE dataset. It could be that this dataset is not as difficult, or that the annotations do not require the model to leverage syntactic information to the same extent, such that the added noise of the method was larger than the potential benefit. For example, in Table 1, it can be seen that Hindi did benefit from the approach in the STANDARD setting. Validating these hypotheses was beyond scope of this work.
> * **[Presentation Improvements 2]** This is due to the technical need to position the tables to fit 8 pages. With the extra page we will change the table numbering.

---

### Official Review · Reviewer_W6Rf · 2023-08-05

**Soundness:** 4

**Excitement:**

4: Strong: This paper deepens the understanding of some phenomenon or lowers the barriers to an existing research direction.

**Missing References:**

Missing a key reference: Aufrant et al. (2016), Zero-resource Dependency Parsing: Boosting Delexicalized Cross-lingual Transfer with Linguistic Knowledge

For section §2 it is appropriate to mention, because a) it has anteriority over Liu et al. (2020a) on the POS-based language model idea, and b) it also includes an instance of the "using WALS" approach but with source-side (training-time) reordering, as this work does (whereas Meng et al. 2019 is at inference time).

But perhaps more importantly, it already provides answers to some of the issues encountered in this work: how reordering can be detrimental, why avoiding deterministic reorderings, etc. See [Re-A] above.

**Paper Topic And Main Contributions:**

This paper presents a new method to improve cross-lingual transfer based on source-side word reordering (operated at the level of dependency subtrees). The authors do so by estimating word order preferences from data, then casting them as constraints to an SMT solver to identify the optimal reordering. The method is experimented on 3 tasks and on typologically diverse languages. Results are ambivalent (usually beneficial but detrimental for some language pairs) but the authors propose a mitigation measure (concatenating original and reordered data) that is successful. Experiments are extended to the few-shot scenario and to comparison among architectures, which yields new insights on the generalizability of the method as well as on properties of existing algorithms.

**Questions For The Authors:**

[Question A] Lines 75-80, I don't understand the point made here. Isn't "use pre-nominal adjectival modification" the same as "high probability for adjectival modifiers to precede the headword"? What difference did the authors want to make here?

[Question B] Line 280 states "assuming that the target language does not have determiners", but it is unclear how this information has been used in the given example (has it?). How is the absence of determiners accounted for in the method? (Side remark related to [Re-A] above: Aufrant et al. also considered the case of absent determiners)

[Question C] Line 300 mentions an approach to estimate POCs without a treebank, but it is not clear where in the paper this is experimented with. For which languages has this method been used, and in which Table are the corresponding results?

[Question D] In the §4.2.1 experiment, what has been done exactly for the Thai data? Because line 390 states "never use the same dataset", but both line 381 and Table 6 mention Thai-PUD. Which one is correct?

[Question E] Line 447 mentions the use of TAC English for train and Trans-TAC for test, but Trans-TAC is already a translation of TAC English, so there may be leaking. Is it the case that the Trans-TAC data used is solely translated from the **test** split of TAC English? Can you clarify and justify why this is sound evaluation?

[Question F] For the few-shot scenario (line 518 and Table 4), is the experiment conducted with the models based on XLM or mT5?

[Question G] Line 1025, how does it help mitigating conflicts to discard the irrelevant constraints? If those constraints are not applicable to that subtree (because root labels do not match), then they do not take part in the computation of the solver, right? So why would they be the ones preventing the solver from finding an ordering?

**Reasons To Accept:**

[Ac-A] The proposed method is thought-enriching and opens new research avenues.

[Ac-B] The authors experiment with a large variety of languages (typologically diverse), tasks, settings (zero-shot, few-shot, different architectures, etc.), which yields more comprehensive and generalizable results.

[Ac-C] Much appreciated the initiative to include a longer discussion on the method's shortcomings in the annex (beyond the "Limitations" paragraph).

[Ac-D] Annexes provide comprehensive experimental details and results, including standard deviations, which is very good methodology.

**Reasons To Reject:**

**[Re-A]** The idea underlying this contribution is quite interesting (using a SMT solver to improve upon prior similar work), but it also seems that the authors missed an important related work (Aufrant et al. 2016, see Missing references), and that if accounting for it they would possibly have made some choices differently, and interpreted some of their results differently.

In particular, Equation (2) shows that they discretize the statistics measured empirically (turning a slight preference into a deterministic hard constraint). Why? Aufrant et al. (2016) have precisely discussed (and implemented accordingly) how it is beneficial to have "*smooth transformations (with mean preference rate objectives and error margins)*", in other words to avoid deterministic word order, because it is a linguistic fact that for some languages the word order preference is not necessarily deterministic for a given label pair. And they also show how deterministic reordering can be detrimental in practice, by losing useful (balanced) information in the source treebank.

Impact:
- [Re-A1] The argument made on line 331 (that ENSEMBLE works better because statistical estimation creates imperfection) appears dubious in light of that prior work. On the contrary: it is because the proposed method overlooks the statistical nature of word order preference, that ENSEMBLE works better. Indeed, when source has mostly prenominal adjectives and the target has a very slight preference for postnominal, then STANDARD contains 100% postnominal, whereas ENSEMBLE contains half prenominal / half postnominal (= much closer to the target)… which is exactly what Aufrant et al. advocated for. This sheds a completely new light on ENSEMBLE, and significantly changes the interpretation of the results.
- [Re-A2] Same for the case of performance decrease in case of close languages: a simple interpretation is that close languages have already similar word order ratios for a given label pair, and the discretized constraints move them further. This is exactly the "French to Italian" case analyzed by Aufrant et al. So these observations may just be an artefact of the choice to discretize the constraints, not an evidence of the applicability of reordering in general depending on the language pair.
- [Re-A3] And for the remark line 506 on MTOP results opposite to Multilingual-TOP ones: since language sets are different, that again may just be an artefact of the fact that target languages on one side or the other have more deterministic or more balanced word order preferences.

The authors acknowledge on line 1039 that there may be an issue with this discretization, but the analysis does not go as far as observing how that impacts (and possibly invalidates) some of their own conclusions and analyses in the paper. Actually, the mention of "statistical validity" (line 1041) looks as if the authors consider that either "nmod < amod" or "amod < nmod" is the appropriate constraint and when measuring P=0.51 it only prevents to identify which one is, NOT that the appropriate one would indeed be "half of each".

**[Re-B]** There is also an issue with the realism of the targeted scenarios. Often when working on low-resourced scenarios, there are irremediable issues that prevent from being fully realistic (e.g. evaluation can only be done when sufficient data exists, so not for actually low-resourced languages). So it is fully OK to assume that such work is done in best-effort mode. But this does not mean discarding the corresponding issues as being irrelevant, but rather to acknowledge them and acknowledge they are not solvable. Plus, care must be taken that the proposed methods would actually be meaningful in a real low-resourced scenario.

More precisely:
- [Re-B1] Regarding the pre-requisite for an UD treebank to estimate the POCs, line 359 accurately observes that UD is available for many languages anyway. However, treebank availability does not mean availability of a treebank with size comparable (hence the same reliability of POC estimation) to the large treebanks used in Table 5 for French, Spanish, or German. So this presumably overestimates a lot the quality of the POCs estimated in actually low-resourced settings. In particular, the treatment made of footnote 6 is questionable: precisely this is a real-world scenario of low-resourced language, so the conclusion "impossible to extract" raises major questions on the applicability of the method to actual low-resourced cases.
- [Re-B2] The scenario for estimating POCs without a treebank also does not seem very convincing. If using annotation projection to produce a treebank in the target language, why only using it for estimating POCs (and then training on a treebank from a different language), rather than directly using the projected trees as training treebank? And same for the other tasks, if there is parallel data to do annotation projection, then isn't it appropriate to project the available TOP & RC annotations through that parallel corpus, instead of resorting to a convoluted approach through another corpus and through data transforms? Or has this issue already been thought of, and were there specific reasons to prefer that approach?

**[Re-C]** Finally there is a number of places where the analysis is too shallow, or sometimes completely missing:
- [Re-C1] For the relation-classification task, lines 524-534 only describe results, without any comment on what can be concluded from those results.
- [Re-C2] The remark line 502 on English vs French overlooks that English and French are indeed related, but have nevertheless a number of important typological differences (pre/postnominal adjectives for instance), so it is unsurprising to observe gains. What is the concept of "typologically distant" used here?
- [Re-C3] Only English is used as source language, which is clearly not neutral when considering typology (English being rather atypical among the 7000 languages, regarding typology). Surely these experiments are computationally expensive, so experimenting with many sources was probably not doable. But the motivations and impact of that choice would have deserved at least a comment.
- [Re-C4] In Table 5, a number of s2s results are so low that it is hard to consider that the model does anything (or at least, it does not outperform a crude heuristic such as "dependency to the left neighbour, with the majority-class as relation label"). This raises many questions on how to interpret the score increase, so this would have at least deserved a comment (more than just calling it "subpar").

**Reproducibility:**

4: Could mostly reproduce the results, but there may be some variation because of sample variance or minor variations in their interpretation of the protocol or method.

**Reviewer Confidence:**

4: Quite sure. I tried to check the important points carefully. It's unlikely, though conceivable, that I missed something that should affect my ratings.

**Typos Grammar Style And Presentation Improvements:**

- Line 56, it is not discussed what "old" and "new" mean here, so it is hard to understand what phenomenon is referred to. This seems very marginal compared to the topic of the paper, so maybe not worth a long discussion, but is it possible to rephrase in a more explicit manner?
- Typo line 107: suggest --> suggests
- Typo lines 167, 221, 251, 259, 267: constrains --> constraints
- Line 261 (and possibly elsewhere), sub-tree --> subtree (to match the spelling in the rest of the paper)
- Typo line 265: on of --> on
- Typo line 256: the the --> the
- Typo line 490: the the --> the
- Typo line 531: then --> than
- Table 2 formatting: it would be useful to display differently the results that are an increase and a decrease (to see more easily which languages are concerned). Use for instance bold, underline, decorate with an asterisk after… at least for Table 2 since this is explicitly discussed at line 474, but if possible for other Tables as well.
- Tables 4 & 5: use the same precision for all numbers (e.g. 67.0 instead of 67 for Fr-300-Base)
- Typo line 587: tends --> trends
- Typo line 594: suggest --> suggests
- Typo line 604: able boost --> able to boost
- Typo line 620: results --> result
- Typo line 628: suggests --> suggest
- Typo line 633: remove "more"
- Line 655, "multiply-parallel" is awkward, usually this is called "multi-parallel"

---

> ### Author Rebuttal · Authors · 2023-08-28
>
> We thank the reviewer for the review, for the reference, and for the suggestions for improvement. We will incorporate them.
> * **[Re-A]** We thank the reviewer for the reference, which we have indeed missed, and agree that the hard constraints imposed by the original formulation of our method can be detrimental for some target languages. We will emphasize this. However, as the reviewer mentions, the requisite smoothing is essentially recovered by combining the pre-processed and the original training data in the Ensemble setting (more precisely, it mitigates the negative effect of imposing wrong constraints relative to only using vanilla English data).
>
>     We disagree that this “significantly changes the interpretation of the results” because the variability in the target word order, of which we are aware and which we will stress more in the revised version, is not the only source of noise that the ensemble setting mitigates: there is also the issue of the conflict between the word order seen by the encoder in pre-training vs. fine-tuning. For example, the models fine tuned on the reordered English almost always obtain lower results on the reordered English validation sets than the vanilla models on the vanilla validation set. Another source is the errors in frequency estimation for rarer dependency-label pairs (notice that ensembling improves the results for almost any type of target language).
>
>
>     We also do not think that knowledge of this work would have materially influenced our methodology as the approach by Aufrant et al. (2016) , while being quite comprehensive, retains some of the common issues we wanted to address: it is local (reorderings are only allowed inside 3-word windows), it is not tree-aware (transformations are defined based on POS sequences, not parent-conditioned dependency labels), and in addition to the statistics of target-language POS tags, it relies on manually-defined rules derived from WALS. Furthermore, as mentioned in the Shortcomings section, we fully agree that this is an important future work avenue, but one that requires substantial additional effort.
>
>
>     _“Actually, the mention of "statistical validity" (line 1041) looks as if the authors consider that either "nmod &lt; amod" or "amod &lt; nmod" is the appropriate constraint and when measuring P=0.51 it only prevents to identify which one is, NOT that the appropriate one would indeed be "half of each"._” —  We meant to acknowledge the issue, without offering concrete solutions.  Furthermore,  “Half of each” is not a constraint, but rather a probabilistic matching of the target word order, and thus the way to model this is not straight-forward (in our method). Furthermore, it ignores lexical effects, information structure, and other sources of WO variability. Generally, our approach in the ensemble setting is based on mixing the prevalent target WO (which we can estimate with some confidence) with the source word order (which we know). We also experimented with setting higher thresholds for hard constraints, i.e. setting them only when P >= 0.75 or 0.8, but this did not lead to improved performance.
>
> * **[Re-A1]** We agree that we need to stress the impact of ensemble setting with targets with soft constraints, but we consider the proposed reasoning to only provide a partial explanation. If the variability in the target order were the dominant factor, ensembling would be markedly damaging for languages with strict deviant word orders, such as Korean or Irish, because it would only add noise by preserving misleading English structures. This, however, is not the case as ensembling almost always yield the best results, with only Japanese Multilingual Top demonstrating a real drop in performance (for UD Korean there is a slight drop, but for RE Korean there is a substantial increase). We agree that we need to investigate this case further and will acknowledge this in the final version.
> * **[Re-A2]** As mentioned above, we think that while discretizing the constraints might be a reason for it, it is not the only one. If it was, then there would be language-pairs for which the ensemble method would be damaging. While Aufrant et al. did point out an important issue (which we acknowledged in the Shortcomings section), they made different design choices (hand-written rules are allowed, they are using WALS, etc.), which are not easily applicable to our work. As mentioned in the Shortcomings section, we agree that this is an important future work avenue.
> * **[Re-A3]** As mentioned above, ensembling to a large extent infuses the training data with the source, i.e. English, word order bias. Therefore if that was the only reason, ensembling should somewhat lower performance gains on languages with _hard WO preferences different from English_, which it often does not. Japanese and Italian have relatively free WOs, so the reversal we are witnessing is indeed surprising. Furthermore, while we agree with the comment and will address it, we personally suspect it to be more a case of peculiarities in the multilingual-TOP dataset.
> * **[Re-B]** We acknowledge this and agree with your statement. We do feel that we were careful in mentioning the limitations of our approach, but we will go over the paper again and make sure to clarify it.
> * **[Re-B1]** We do not require a treebank of the size comparable to that of French, Spanish, or German with tens of thousands of annotated sentences. For example, the PUD treebanks we used contain only 1000 sentences. The  “impossible to extract” case that we mention contains a mere 17 examples. We will add a clearer mention of the PUD size so it is clearer, and we acknowledge that experimenting with even lower sizes of treebank is an important avenue for future work.
> *  **[Re-B2]** We mentioned the approach as a possible extension, relying on the fact that RC19 used this approach successfully with their reordering algorithm on top of annotation projection (on UD). While their setting is different, as they did not use modern-multilingual models (and thus we cannot make direct conclusions from their results), we expect to see similar gains. This is because the texts to which we can more or less reliably project UD annotations, are restricted to the New Testament, which is not lexically representative. Additionally, annotation projection introduces its own kind of noise , which is different from the one reordering algorithms introduce. We will address this in the final version.
> * **[Re-C1]** We feel these results are largely in line with our other observations and did not find anything insightful to add.
> * **[Re-C2]** Despite some differences in grammar, English and French possess many typological commonalities, which makes them both parts of the Standard Average European linguistic area (see the discussion and references in https://en.wikipedia.org/wiki/Standard_Average_European). Past work often did not see improvements for European languages such as French and often saw a decrease in performance. In our work too, French did not see gains in UD parsing, and the same can be observed for the application of RC19 to the MTOP mT5 model. So we were indeed surprised by the mentioned gains.
> * **[Re-C3]** Experimenting with more source languages is indeed impractical. The choice of English as a source language was motivated by the fact that most datasets are in English, and most multilingual models had more English training data. As some related work (e.g., Nikolaev & Pado, Word-order Typology in Multilingual BERT, SigTyp 2022) shows, replacing English with another high resource source language that has a more flexible word order and richer morphology (such as Czech or Russian) can lead to some improvements in cross-lingual transfer, but in practice those are rather minor. We touched upon it in the introduction, and did not mention it further as using English as the source language is the standard setting in all past works. We will add a comment on this and clarify the motivation further.
> * **[Re-C4]** In this study, we did not attempt to explain the unsatisfactory performance of base models but to initiate discussion on the issue. Our hypothesis that s2s models fail to project to diverging word orders seems to be well supported by Table 5, and while the exact reason for this is a fascinating research problem, we leave it for future work.
> * **[Question A]** Here we restate the constraint usually formulated in terms of POS (Adj–Noun vs. Noun–Adj) in terms of parent-aware dependency relations: amod–nsubj/obj/obl, etc. vs. the other way around. We will clarify this point.
> * **[Question B]** The absence of determiners in the target language allowed us to leave the position of determiners on the source side unrestricted. In the example, we kept them in place for presentational clarity, but their position is largely irrelevant.
>  \“Side remark related to **[Re-A]** above/: Aufrant et al. also considered the case of absent determiners” — yes, they removed them, and elements of other types, when appropriate, based on information from WALS. We do not use WALS and restrict ourselves to using subtree preserving reorderings. (We note that the definition of a determiner in UD and WALS can differ, even across corpora. For example, deictic pronouns can be treated as determiners or amod’s in languages without articles, such as Japanese, which makes blanket removal of word categories dangerous.)
> * **[Question C]** We only mentioned the approach as a possible extension, relying on the fact that RC19 used this approach successfully with their reordering algorithm. We will make this clearer. As a side note, we did experiment with this approach, and saw some success, but a comprehensive study of this setting involves retraining all of the reported models for all languages and experimental settings, which is too computationally demanding.
> * **[Question D]** You are correct, this is a mistake on our end. Thanks for catching it. Thai has a single UD treebank and this is the source of the mistake. We will clarify that or split the treebank and re-run the experiments if time permits (this will also allow us to report results on an even smaller treebank than 1000 sentences).
> * **[Question E]** We only used a subset of the TACRED training set (as mentioned in the paper), which does not include the Trans-TACRED examples. We will emphasize that. In any case, the vanilla and the reordered models are trained exactly the same way and with the same data, so the comparison is valid.
> * **[Question F]** mT5. Thank you for noticing this omission
> * **[Question G]** We had a choice between solving for a universal ordering of each subtree type (by label) once, assigning positions to all possible children, and repeating the analysis on a case-to-case basis. The second option boils down to not including irrelevant constraints on a sub-tree basis, which is essentially a mitigation because, e.g., a CFG, even a probabilistic one, would assign a recommended total ordering to all subtrees, thus increasing the number of subtree types that have ordering conflicts. For example, in a specific language, we might have a constraint of amod->nmod, for a subtree of type “subj”, which contradicts another constraint under the same subtree. If in the specific subtree of label “subj” we are reordering, amod does not appear, we will remove this constraint. This way we are removing irrelevant constraints that might contradict one another.

---

### Meta-Review · Area_Chair_Xa9X · 2023-09-06

**Recommendation:** 2

**Metareview:**

The authors propose an interesting and promising approach, but the reviewers have identified several points that need to be improved before this work can be published: quantification of similarity between languages, analyses that are not sufficiently detailed, etc.

---

### Decision · Program_Chairs · 2023-10-07

**Decision:**

Accept-Findings

**Comment:**

The authors propose an interesting and promising approach, but the reviewers have identified several points that need to be improved before this work can be published: quantification of similarity between languages, analyses that are not sufficiently detailed, etc.